# Alterations in Nonvolatile Components of Tea (*Camellia sinensis*) Induced by Insect Feeding under Field Conditions

**Ayumi Ito [1], Jin Kamiya [2], Nakako Katsuno [1,3] and Emiko Yanase [1,3,\*]**

[1] The United Graduate School of Agricultural Science, Gifu University, 1-1 Yanagido, Gifu City 501-1193, Japan
[2] Gifu Prefectural Agricultural Technology Center, 729-1 Matamaru, Gifu City 501-1152, Japan
[3] Faculty of Applied Biological Sciences, Gifu University, 1-1 Yanagido, Gifu City 501-1193, Japan
\* Correspondence: yanase.emiko.h2@f.gifu-u.ac.jp; Tel./Fax: +81-58-293-2914

**Abstract:** Tea leaf components are affected by environmental factors such as insect feeding, and metabolites have been studied using specific insect and tea leaf model systems. However, in gardens, tea leaves are eaten by various insects. Because the components of tea may affect human health, the effect of insect damage on metabolites needs to be clarified. The aim of this study was to investigate the effects of insect feeding on the nonvolatile components of tea in a field experiment. Furthermore, we determined the effects of insect damage on the composition of black tea products. Tea leaves were sampled from insect-attacked and moderately insect-attacked areas. Catechins were quantified by ultra-performance liquid chromatography. Tea leaves were analyzed by ultra-performance liquid chromatography–mass spectrometry, followed by orthogonal partial least squares-discriminant analysis and molecular networking analysis. The nonvolatile components in insect-attacked leaves were significantly affected. The amounts of gallate-type catechins in highly attacked leaves were approximately 1.2 times higher than in moderately attacked leaves. Furthermore, highly attacked leaves had increased levels of afzelechin gallate, procyanidins, and hydrolyzable tannins. These results varied with previous reports that used model systems with specific insects. In addition, some of these compounds were also detected as characteristic components in black tea from highly attacked leaves.

**Keywords:** insect feeding; molecular networking analysis; orthogonal partial least squares-discriminant analysis; nonvolatile components; polyphenols

## 1. Introduction

Tea is a popular beverage consumed worldwide, second only to coffee. In recent years, the various functionalities attributed to catechins in tea, such as antioxidant and antitumor activities, have attracted attention [1–6]. Therefore, tea is consumed as a functional food [7,8]. Catechin content in tea leaves is reported to be around ~30%; however, catechin content and composition are not constant in tea leaves. They vary with the season, generally being lowest in the first-flush and increasing through the third-flush tea [9]. Furthermore, the plant's secondary metabolites, including tea catechins, are generally affected by environmental factors, such as cultivars, soil, sunlight, and temperature [10]. In order to properly evaluate the health benefits of daily tea consumption, the variation of tea components due to environmental factors needs to be clarified.

Insect feeding affects tea yield and quality and is a common problem in tea gardens [11–13]. There are two major types of pest insects, chewing (such as tea leaf rollers and tea tortrix) and piercing–sucking herbivores (such as spider mites and tea leafhoppers), and agrochemicals are used to inhibit their activity. Many studies have indicated that insect feeding induces unique volatile components in tea leaves [11,14–16]. This phenomenon is known as the plant defense response and is triggered by physical destruction or exposure to components of oral secretions or oviposition fluids [11]. For example, leafhopper feeding

causes an increase in linalool, geraniol, and diendiol I. These volatile components induced by leafhopper feeding contributes to the flavor of "Oriental Beauty", an oolong tea [16,17].

Nonvolatile components, such as polyphenols and alkaloids, contribute to the plant's defense [18]. The levels of nonvolatile metabolites have been studied using model systems in which leafhoppers were introduced to potted tea plants in mesh bags. The results showed that insect feeding resulted in a decrease [19] or no significant change [12] in catechins, while theaflavins (TFs) increased [12]. However, in actual gardens, tea leaves are affected by various insects; thus, the changes in their metabolites do not always conform with those of model systems. Therefore, the influence of insect damage on tea leaf components should be investigated under field conditions.

The aim of this study was to investigate the influence of insect damage on nonvolatile components under field conditions of insect-attacked and moderately insect-attacked areas. Multivariate and molecular network analysis facilitated the detection of nonvolatile components that contributed to changes induced by insect damage. Furthermore, the changes in the components of fresh tea leaves may affect the quality of black tea products. Therefore, the composition of black tea products from insect-attacked leaves and moderately insect-attacked leaves were also studied.

## 2. Materials and Methods

### 2.1. Materials

Epigallocatechin-3-O-gallate (EGCG) was a kind gift from Nagara Science Co. (Gifu, Japan). Epigallocatechin (EGC) was prepared from EGCG by enzymatic hydrolysis. TFs were synthesized as previously reported [20]. Epicatechin and sulfisoxazole were purchased from Tokyo Chemical Industry (Tokyo, Japan), and caffeine from Nacalai Tesque (Kyoto, Japan).

### 2.2. Field Setting

Insect-attacked and moderately insect-attacked areas were set in a tea field at the Ikeda Cho Chagyo Center (the Tea Industry Center of Ibi Tea, Gifu Prefecture, Japan). Each area had five ridges (width: 1.8 m, length: 22–26.7 m/ridge), and one ridge was left unused between areas. The level of insect attack was controlled using pesticides. In the moderately insect-attacked area (hereafter area A), insects were controlled with pesticides, while in the insect-attacked area (hereafter area B), no pesticides were used for approximately one month before the first tea was picked. The lower number of insect attacks in area A compared to area B was confirmed by the number of *Empoasca onukii* Matsuda counted using the knock-off method (explained in Section 2.3).

### 2.3. Monitoring the Number of Insects

The data reported by the Gifu Ken Byogaityu Bozyo Center (the pest control center of Gifu Prefecture, Gifu, Japan) and Ibi Region Agriculture and Forestry Office (Gifu Prefecture, Japan) were used [21]. The pheromone trap and knock-off methods were used to monitor insects. Sticky-type traps with adhesive boards were fixed on tea trees to predict pest outbreaks. Lures were placed in the center of the board. The insects captured (*Adoxophyes honmai* Yasuda, *Homona magnanima* Diakonoff, and *Caloptilia theivora* Walsingham) were counted every five days. Knocking-off was performed 10 times/half ridge to 182 mm $\times$ 257 mm boards sprayed with adhesive spray on three ridges in each area. The number of *Empoasca onukii* Matsuda was counted and the average count was calculated.

### 2.4. Tea Sample Preparation

Tea leaves (*Camellia sinensis* L. cv. Yabukita) were obtained from the Ikeda Cho Chagyo Center (the Tea Industry Center of Ibi Tea, Gifu Prefecture, Japan) in April, June, and August 2022. Sample preparation was performed in the same manner as in our previous report [22]. The lyophilized (Tokyo Rikakikai Co., Ltd., Tokyo, Japan) samples were ground

to a powder. Before analysis, distilled water (0.2 mL), 33% chloroform in methanol (0.6 mL), and 0.1 mg/mL sulfisoxazole (Tokyo Chemical Industry, Tokyo, Japan) in methanol (10 µL, as an internal standard) were added to each sample (15 mg) and mixed using a vortex mixer (Vortex Genie 2; Scientific Industries, Bohemia, NY, USA). After 1 h, distilled water (0.2 mL) and chloroform (0.2 mL) were added, and the resulting mixture was centrifuged (4043 rpm, 5 min). Quantification of catechins and TFs and ultra-performance liquid chromatography–mass spectrometry (UPLC–MS) were then performed using the supernatant.

The lower phase (0.2 mL) was collected in an Eppendorf tube and concentrated using a centrifugal concentrator. It was redissolved in methanol (0.2 mL) for caffeine quantification.

Processing of black tea from fresh tea leaves was performed as follows: four kilograms of fresh tea leaves were processed into black tea by withering for 17.5 h (until the moisture content reached ~65 wt. %), mechanical rolling for 1 h (Kawasaki Kiko, Co., Ltd., Shizuoka, Japan), fermentation for 1 h, and drying using a dryer (Terada Seisakusho Co. Ltd., Shizuoka, Japan) for 45 min (chamber temperature at 80 °C).

## 2.5. Quantification Analysis

Quantification of catechins, TFs, and caffeine was performed based on previous reports [22] with slight modifications. A UPLC system (Waters H-class; Waters Corp., Milford, MA, USA) equipped with an Acquity UPLC BEH C18 column (2.1 mm I.D. × 100 mm, 1.7 µm; Waters Corp.) was used for chromatographic separation and analysis. The column and analysis temperatures were 35 °C, and the sample injection volume (auto-sampler) was 1 µL. UV monitoring was conducted at 280 nm, and flow rate was set at 0.4 mL/min. The mobile phases for catechin quantification were 1% formic acid in water (A) and methanol (B), and the following gradient was used: 0–5 min, 5% B; 19 min, 15% B; 23 min, 25% B. The mobile phases for TFs quantification were 1% formic acid in water (A) and acetonitrile (B), and the gradient was applied as follows: 0 min, 15% B; 4–11 min, 22% B; and 11.1–13 min, 100% B. The mobile phase for caffeine quantification was 0.5% formic acid in 10% acetonitrile/$H_2O$. Quantitative analysis was performed using calibration curves for catechins, TFs, and caffeine (Figure S1) with authentic standards or synthetic samples [20]. Tukey's honest significant difference test for catechins, and Student's *t*-test for TFs and caffeine were used to detect differences between groups.

## 2.6. Ultra-Performance Liquid Chromatography–Mass Spectrometry (UPLC–MS) Analysis

UPLC–MS analyses of tea extracts were performed as previously reported [22]. The UPLC system was coupled to a quadrupole time-of-flight mass spectrometry (Xevo G2 QTOF; Waters Corp.) operated under electrospray ionization conditions at a mass resolution of 20,000 and controlled by MassLynx 4.1 (Waters Corp.) software. The collision energy was 6 V, and source parameters were: capillary voltage, 2.5 kV; sampling cone voltage, 30 V; extraction cone voltage, 4 V; source temperature, 150 °C; desolvation temperature, 500 °C; desolvation gas flow, 1000 L/h; and cone gas flow, 50 L/h. Sodium formate (0.5 mM) was used as the calibration standard. Leucine Enkephalin (2 µg/mL, *m/z* 556.2771 in positive mode) was used as the lock spray at a 10 µL/min flow rate. Samples were measured in positive mode. An Acquity UPLC BEH C18 column (2.1 mm I.D. × 100 mm, 1.7 µm; Waters Corp.) was used for chromatographic separation. The column temperature was 35 °C, and the sample injection volume (autosampler) was 1 µL. The mobile phases (flow rate: 0.4 mL/min) were 1% formic acid in water (A) and methanol (B), and the following gradient was applied: 0 min, 5% B; 8 min, 15% B; 11 min, 25% B; 13 min, 32% B; 20 min, 40% B; 26 min, 55% B; and 32–34 min, 95% B.

For the survey scan, scans were performed between *m/z* 100–1500, with a switching intensity threshold of 3000 counts. MS/MS data of the selected precursor ions were acquired between *m/z* 100 and 1500, with a maximum of three precursor ions. The collision energy was set at 15–35 eV.

*2.7. Data Analysis*

Mass detection and alignment were performed using MZmine 2.37 [23]. Parameters were set as follows; mass detection: 100 (MS scans), 4 (MS/MS scans); ADAP chromatogram builder: minimum group size 4, group intensity 100, minimum height 300, $m/z$ tolerance 0.02 Da; local minimum search deconvolution algorithm: threshold 20%, search minimum retention time (RT) range 0.03, median $m/z$ center calculation; isotope features: 0.02 Da mass tolerance, 0.5 min RT tolerance; feature alignment: $m/z$ tolerance 0.02 Da, RT tolerance 0.5 min; metaCorrelate: RT tolerance 0.5 min, minimum height 300, noise level 100; Pearson correlation: 0.85.

Data processing and normalization were performed using MetaboAnalyst 5.0 [24]. All data were row-wise normalized, Pareto-scaled, and log-transformed to perform principal component analysis (PCA) and orthogonal partial least squares-discriminant analysis (OPLS-DA).

*2.8. Molecular Networking Analysis*

A molecular network was created with the Feature-Based Molecular Networking workflow [25] on Global Natural Product Social Molecular Networking (GNPS; https://gnps.ucsd.edu [26]). Mass spectrometry data were first processed with MZmine 2.37, and the results were exported to GNPS for Feature-Based Molecular Networking analysis. The data were filtered by removing all MS/MS fragment ions within $+/-$ 17 Da of the precursor $m/z$. MS/MS spectra were window-filtered by choosing only the top six fragment ions in the $+/-$ 50 Da window throughout the spectrum. The precursor ion mass tolerance and the MS/MS fragment ion tolerance were set to 0.02 Da. A molecular network was created where edges were filtered to have a cosine score above 0.7 and more than 5 matched peaks. Edges between two nodes were kept if each node appeared in the other's top 10 most similar nodes. Finally, the maximum size of a molecular family was set to 100, and the lowest-scoring edges were removed until the size was below this threshold. The analog search mode was used by searching against MS/MS spectra with a maximum difference of 100.0 in the precursor ion value. The library spectra were filtered in the same manner as the input data. All matches kept between the network and library spectra were required to have a score above 0.7 and at least 4 matched peaks. DEREPLICATOR annotated the MS/MS spectra [27], and the researcher provided additional edges. The molecular networks were visualized using Cytoscape 3.9.1 software [28].

*2.9. Color Difference*

Distilled water (10 mL) was added to each sample (20 mg) and extracted at 95 °C for 10 min. The extract was filtered and cooled to room temperature. The extract (7 mL) was diluted with water (4.5 mL). This solution was placed into a cell, and the L* (lightness), a* (redness), and b* (yellowness) values were measured using a color-difference meter (Konica Minolta Inc., Tokyo, Japan). Measurements were carried out in triplicate. The color difference ΔE was calculated as per the following equation:

$$\Delta\text{E} = \sqrt{(\Delta\text{L}^*)^2 + (\Delta\text{a}^*)^2 + (\Delta\text{b}^*)^2} \tag{1}$$

*2.10. Statistical Analysis*

The significant differences of the quantities of caffeine and TFs were compared by using Student's *t*-test. The significant differences of catechin levels were estimated through Tukey's HSD test. Statistical significance was set at $p < 0.05$.

## 3. Results

*3.1. Number of Pest Insects*

The pest insect infestation data published by Gifu Prefecture [21] exhibited the same trend as the previous year (Figure 1). The number of *A. honmai* Yasuda increased in early May and late June. *H. magnanima* Diakonoff increased in mid-April and did not

increase significantly afterward. *C. theivora* (Walsingham) increased in early June and slightly in mid-July. *E. onukii* Matsuda increased in mid-June, after which it did not increase significantly.

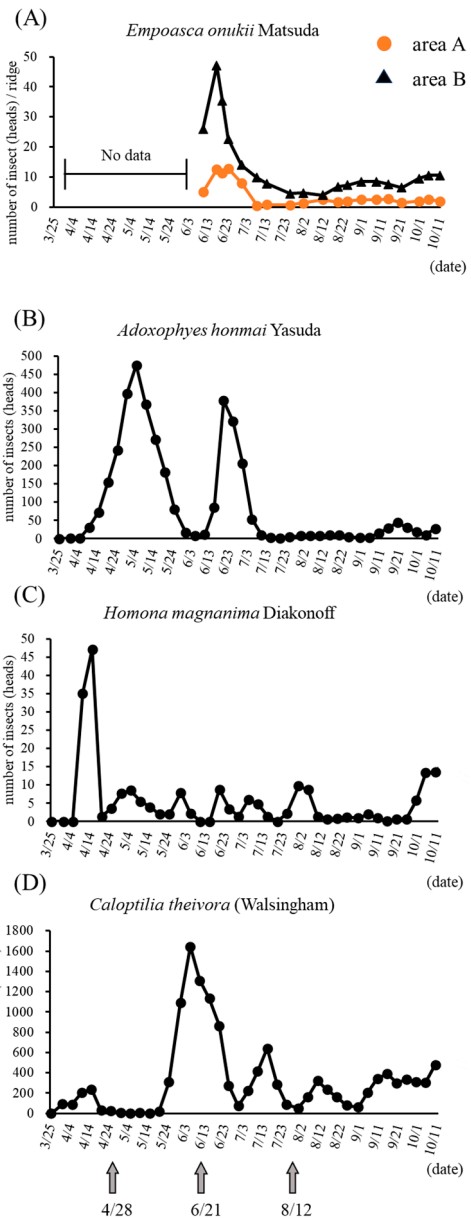

**Figure 1.** The numbers of (**A**) *Empoasca onukii* Matsuda, (**B**) *Adoxophyes honmai* Yasuda, (**C**) *Homona magnanima* Diakonoff, and (**D**) *Caloptilia theivora* (Walsingham) on tea leaves. *E. onukii* Matsuda was counted using the knock-off method. *A. honmai* Yasuda, *H. magnanima* Diakonoff, and *C. theivora* (Walsingham) were counted using the pheromone trap method.

### 3.2. Quantification of Catechins, TFs, and Caffeine

Quantitative analysis was performed to compare changes in the catechin content of tea leaves between areas A and B (Figure 2). In area A, second-flush tea leaves had a higher level of catechins than first-flush tea leaves. Although it has been reported that catechin levels increase with season [9], second-flush tea leaves had the same level as third-flush tea leaves (Figure 2A). No significant differences were observed in total catechins between areas A and B in any harvest period. However, differences were observed in the composition of second-flush leaves. The ratio of gallate-type catechins (ECG, EGCG) was higher, and the ratio of EGC was lower in area B than in area A. While a difference was observed in

the catechin composition of second-flush leaves from each area, it was not observed for first- and third-flush tea leaves. TFs are oxidative catechin dimers formed by polyphenol oxidase. Leafhopper feeding has been reported to increase TF levels [12]. However, for fresh second-flush leaves, no difference in TF quantity was observed (Figure 2B). The quantities of caffeine in second-flush fresh tea leaves were 135.96 and 164.78 µmol/g dry weight (DW) in areas A and B, respectively (~1.2 times higher) (Figure 2C).

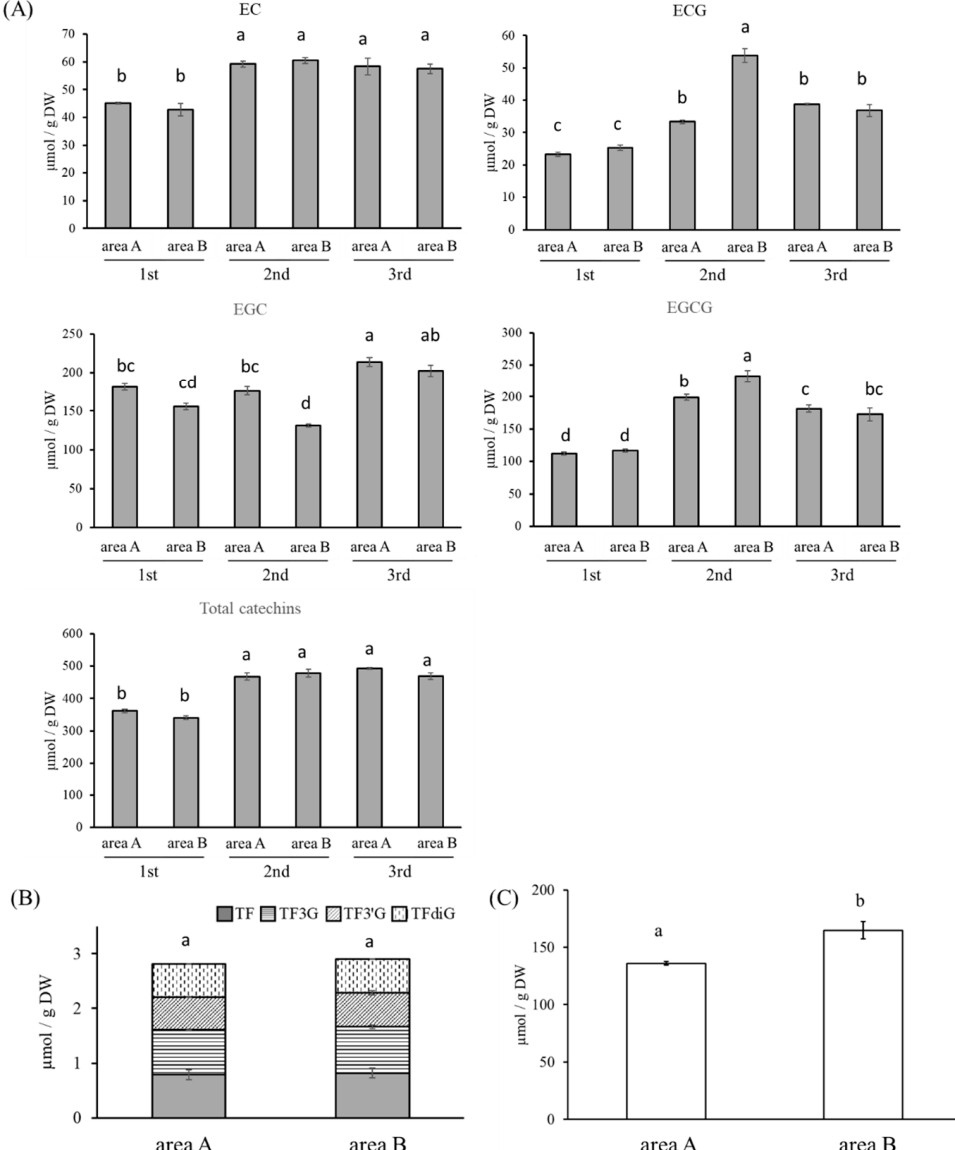

**Figure 2.** Concentration profiles of (**A**) catechins, (**B**) theaflavins, and (**C**) caffeine in second-flush fresh leaves (mean ± SD, *n* = 3). Means in each column for each compound labeled with the same letter are not significantly different (*p* > 0.05).

### 3.3. OPLS-DA of Second-Flush Fresh Leaves

To investigate the effects of insects feeding on components other than those mentioned above, non-targeted analysis by UPLC–MS was performed for the second-flush tea leaves that were affected by insect feeding. Multivariate analysis was performed on the peak area of the detected compounds in the LC-MS analysis to evaluate the influence of insects on nonvolatile components. On OPLS-DA, samples of area B were separated clearly from area A and arranged on the negative side (Figure 3).

Components with the variable importance in the projection (VIP) score greater than 1 and *P* (corr) greater than |0.7| were extracted to find the components that differed between

areas A and B. As a result, 110 components were extracted, of which 84 were abundant in area B (*P* (corr) $\leq -0.7$) (Tables 1 and S2), and 26 were abundant in area A (*P* (corr) $\geq 0.7$). An extracted ion chromatogram (EIC) of the abundant components in area B was prepared and the peak areas were compared. The results showed that these were 1.2–4.9 times larger than those in area A. These components included ECG and caffeine, whose peak areas in area B were 1.4 and 1.2 times larger than in area A.

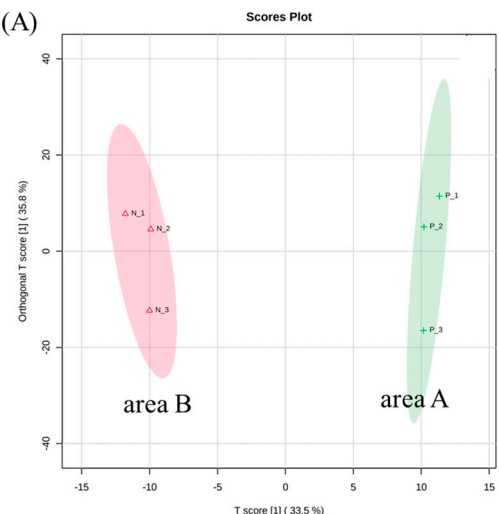

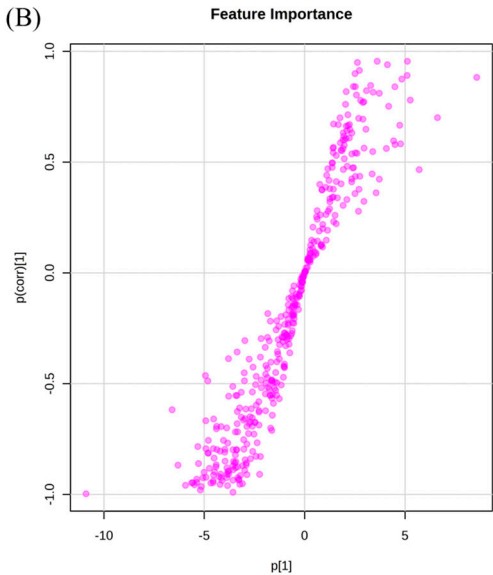

**Figure 3.** OPLS−DA based on the UPLC−MS spectra of the tea leaf extracts. (**A**) The OPLS−DA score plot, and (**B**) the S−plot.

**Table 1.** Components detected by OPLS-DA. RT: retention time.

| ID | RT | Parent Mass *m/z* | MS2 Fragmentation Ion | Dereplication Results | Exact Mass *m/z* | Δppm |
|----|-----|-------------------|-----------------------|-----------------------|------------------|------|
| 7 | 12.23 | 443.0981 | 291, 273, 153, 139, 123 | Epicatechin gallate | 443.0973 | 1.87 |
| 22 | 12.23 | 273.0759 | 147, 123 | N/A | | |
| 99 | 13.72 | 449.0848 | 303, 279 | N/A | | |
| 145 | 13.77 | 427.1021 | 275, 257, 153, 139, 107 | N/A | | |
| 171 | 13.79 | 257.0815 | 139, 131, 107 | N/A | | |
| 176 | 14.03 | 289.0703 | 205, 179, 163, 151, 139 | N/A | | |
| 244 | 7.76 | 286.1283 | 271, 201, 163, 135, 127, 123 | N/A | | |
| 109 | 9.90 | 731.1571 | 579, 443, 427, 409, 289 | N/A | | |
| 158 | 10.85 | 715.1650 | 579, 443, 393, 273 | N/A | | |

**Table 1.** *Cont.*

| ID | RT | Parent Mass *m/z* | MS2 Fragmentation Ion | Dereplication Results | Exact Mass *m/z* | Δppm |
|---|---|---|---|---|---|---|
| 163 | 7.39 | 747.1565 | 595, 459, 427, 409, 289 | N/A | | |
| 195 | 13.12 | 579.1499 | 427, 409, 291, 287 | Procyanidin B2 | 579.1497 | 0.34 |
| 226 | 2.37 | 611.1401 | 443, 425, 307 | N/A | | |
| 239 | 9.69 | 921.1475 | 751 | N/A | | |
| 242 | 9.54 | 899.1673 | 729, 579, 457, 443, 425, 409 | N/A | | |
| 243 | 7.20 | 915.1590 | 745, 595, 459, 457, 425 | N/A | | |
| 334 | 12.09 | 867.1735 | 697, 547, 443, 425, 393, | N/A | | |
| 80 | 15.97 | 355.1733 | 203 | p-Menth-1-ene-4,7-diol 4-glucoside [a] | 355.1727 | 1.72 |
| 154 | 24.56 | 471.2206 | 335, 333 | Linalyl beta-vicianoside [a] | 471.2201 | 1.08 |
| 61 | 5.48 | 185.0414 | 153, 126, 125, 107 | Methyl gallate | 185.0444 | −16.23 |
| 1 | 7.97 | 195.0855 | 138, 110 | Caffeine | 195.0877 | −10.91 |
| 131 | 11.92 | 659.0836 | 489 | N/A | | |

[a]: Annotated as a Na adduct. N/A: indicates not applicable. These IDs weren't annotated by reference spectral libraries.

### 3.4. Molecular Networking (MN) Analysis

The GNPS platform [26] supports the classification of components found in complex mixtures, such as plant extracts, using a large amount of data from mass spectrometry. The components in the resulting network can be annotated by matching to reference spectral libraries. Ion identity MN in the GNPS MN platform [29] was used for profiling chemical changes between areas A and B. Seventy-three components were annotated by matching to reference spectral libraries. Six components (ECG, linalyl glycoside, monoterpenoid glycoside, methyl gallate, procyanidin, and caffeine) were identified as abundant in area B by OPLS-DA (as mentioned in Section 3.3).

The constructed MN from the MS/MS data of the examined accessions comprised 403 nodes connected in 40 clusters (a minimum of two connected nodes) (Figure 4). For MN analysis, the two large clusters consisting of 64 and 59 nodes were formed. The largest cluster (MN I), including 64 nodes, consisted of flavonols and their glycosides; and the second largest cluster (MN II), including 59 nodes, consisted of flavan-3-ols. Moreover, there were clusters of hydrolyzable tannin and volatile glycosides as characteristic clusters.

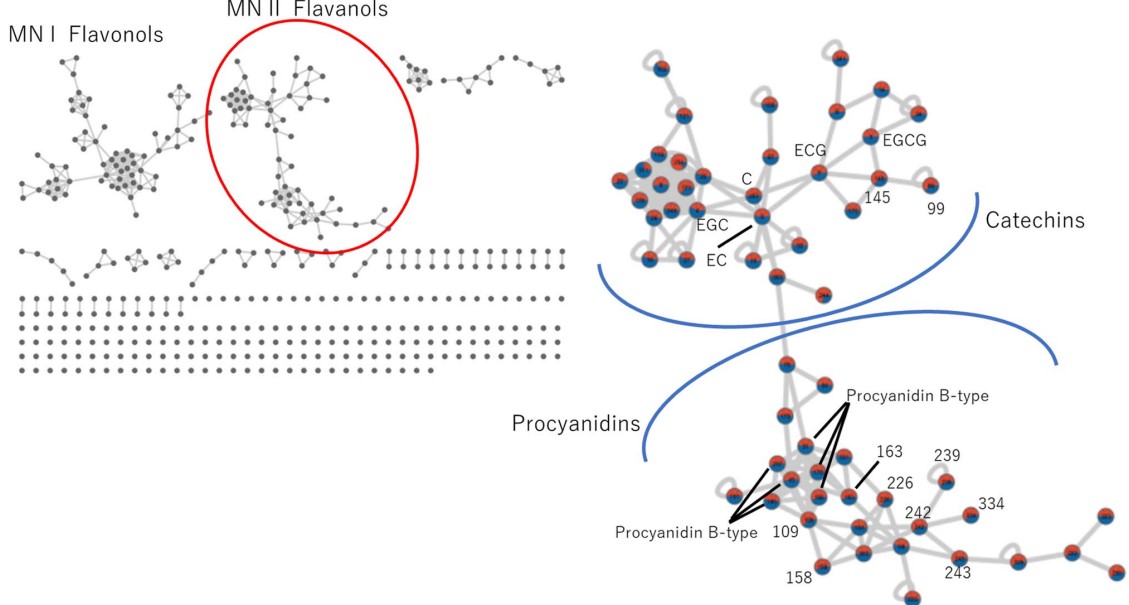

**Figure 4.** Molecular network of components from tea extracts. The different colors of sections in the nodes represent different samples; blue represents area A, and red represents area B. In total, the MN I cluster of flavonols comprised 64 nodes, and the MN II cluster of flavanols comprised 54 nodes. The figure on the right is an enlarged MN II cluster.

### 3.4.1. Catechins and Procyanidin

Catechins and procyanidins constituted the MN II cluster. Catechin, EC, EGC, ECG, and EGCG were annotated, while ID 99 and ID 145 were unannotated nodes in the MN II cluster abundant in area B. In MN analysis, a network of nodes (compounds) is formed based on the principle that similar structures tend to form similar fragment patterns in MS/MS analysis [26]. Thus, connected nodes have similar structures. ID 145 was detected at $m/z$ 427 as a precursor ion and directly connected to EGCG and ECG nodes. This component was 16 Da smaller than ECG and 32 Da smaller than EGCG. Therefore, this component was presumed to be epiafzelechin 3-O-gallate (detected as a proton adduct ion; $[M + H]^+$), with one hydroxy group at the B ring. Moreover, ID 99 was presumed to be the $Na^+$ adduct ion of ID 145. The EIC of this compound showed that it was 1.7 times more abundant in area B than in A.

Six nodes with $m/z$ 579 were annotated as procyanidin B2, a catechol-type catechin dimer with a C4–C8 interflavan bond. In addition to procyanidin B2, there are procyanidin B3 and B4 in tea [30], which are stereoisomers. Moreover, procyanidins with C4–C6 bonds also exist, and the six nodes annotated as procyanidin B2 can be considered stereoisomers or regioisomers. Furthermore, this cluster contained eight components abundant in area B but were not annotated. Tea has been reported to contain not only catechol-type procyanidin dimers, but also procyanidin dimers which consist of pyrogallol-type or gallate-type catechins [30,31]. These non-annotated nodes were presumed to be procyanidins, which consist of different catechins (gallocatechins, afzelechins, gallate-type catechins) (Table 2).

**Table 2.** Procyanidin annotations.

| ID | RT | $m/z$ | Predicted Annotation |
|----|----|----|----|
| 226 | 2.37 | 611.1401 | prodelphidin |
| 109 | 9.90 | 731.1571 | C-CG dimer |
| 163 | 7.39 | 747.1565 | C-GCG dimer |
| 242 | 9.54 | 899.1673 | GCG-CG dimer |
| 239 | 9.69 | 921.1475 | Na adduct of ID242 |
| 243 | 7.20 | 915.1590 | GCG-GCG dimer |
| 158 | 10.85 | 715.1650 | afzelechin-CG dimer |
| 334 | 12.09 | 867.1735 | afzelechin gallate-CG dimer |

### 3.4.2. Hydrolyzable Tannin

Node ID 131, abundant in area B, was connected to node ID 120, which was annotated as hydrolyzable tannin, a hexose combined with two gallate groups ($m/z$ 507). ID 131 ($m/z$ 659) was 152 Da larger than ID 120, suggesting the presence of an additional gallate group. Therefore, it was presumed that ID 131 was a hydrolyzable tannin, a hexose combined with three gallate groups. Although the hydrolyzable tannins were only present in small amounts [32], their level in area B was 2.3 times higher than in A.

### 3.4.3. Volatile Compound Glycoside

ID 80 and 154 were annotated as monoterpene glycosides. These components were 2.6 and 1.6 times more abundant in area B than in area A.

### 3.5. Influence on Black Tea Processing

During black tea processing, polyphenols, amino acids, and sugars in fresh leaves undergo chemical changes due to enzyme reactions [22,33–35]. Consequently, black tea has a unique color, aroma, and taste. Therefore, differences in the components of fresh leaves may affect the components of the black tea product. Non-targeted analysis by UPLC–MS and multivariate analysis were performed for the fresh and black tea samples. According to PCA, the total variance was 81.8%, where PC1 and PC2 accounted for 74.7 and 7.1% of the total variation, respectively (Figure S4). PC1 showed a clear separation of fresh leaves and black tea products. Furthermore, PC2 showed a separation by treatment; samples of

area B were placed in the positive direction of PC2, while samples of area A were placed in the negative direction of PC2.

Moreover, OPLS-DA was performed to investigate the differences in the components of black tea products (Figure S5). Components with a VIP greater than 1 and $P$ (corr) $\leq -0.7$ were extracted to find components rich in area B. As a result, 72 components were extracted. These included afzelechin 3-O-gallate (ID 145), procyanidin (ID 109), and monoterpene glycoside (ID 80), as mentioned above (Table S3).

Since TFs are related to black tea quality, a quantification analysis was performed. The total TFs content in area B was 1.1 times higher than in area A (Figure 5). Theaflavin-3-O-gallate and theaflavin-3,3′-O-gallate contents in area B were 1.3 and 1.5 times higher than in area A.

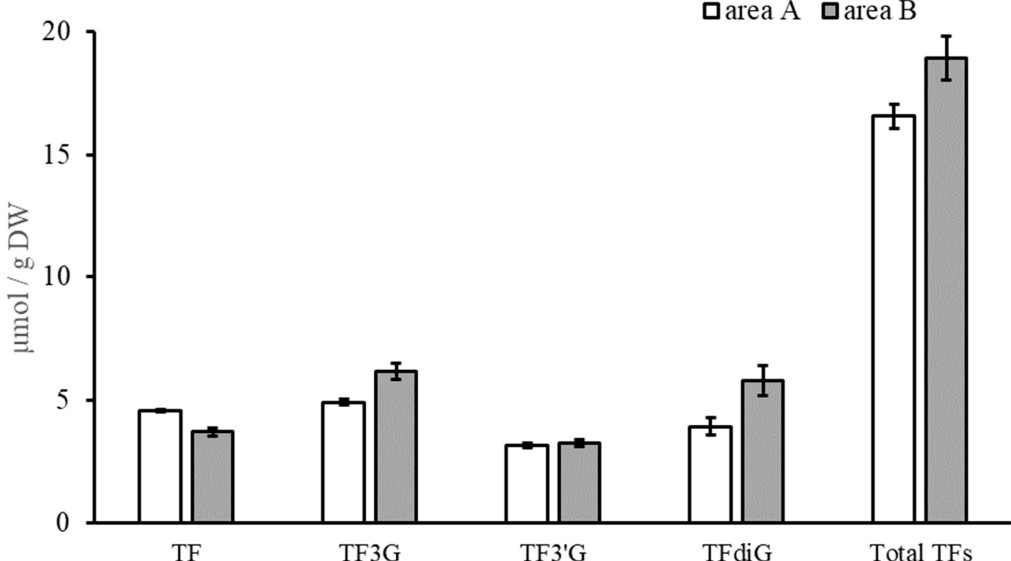

**Figure 5.** Concentration profiles of the theaflavins (TFs) in black tea leaves (mean $\pm$ SD, $n = 3$). TF3G: theaflavin-3-gallate; TF3′G: theaflavin-3′-gallate; TFdiG: theaflavin-3,3′-gallate.

Black tea in area B had higher a* (red–green) and b* (blue–yellow) values, and lower L* (lightness) values than those in area A (Table 3). Furthermore, the color difference ΔE of each black tea was 10.48, indicating different colors. Therefore, the color of black tea products from area B was darker and more orange than that in area A.

**Table 3.** L*a*b* value of each treatment area.

| Sample | L* | a* | b* | ΔE |
|---|---|---|---|---|
| area A | 78.48 ± 0.24 | 11.47 ± 0.14 | 76.57 ± 0.49 | — |
| area B | 70.62 ± 0.21 | 17.36 ± 0.12 | 80.24 ± 0.38 | 10.48 |

Note: Measurements were carried out in triplicate. Data presented are mean ± standard deviation. L*: lightness, a* and b*: chromaticity

## 4. Discussion

The number of pest insects was reported by the Gifu Ken Byogaityu Bozyo Center (Figure 1B–D), which shows trends in the occurrence of insect pests over the whole field. An increasing trend in the numbers of *H. magnanima* Diakonoff was not observed in July 2022 due to high rainfall over that period, but occurrence of the pest insect was generally consistent with previous years [21]. The increasing trend in the number of insects may be due to seasonal variation. Our data suggested that insect feeding affected the second and third-flush tea leaves more than the first. Figure 1A shows the number of *E. onukii* Matsuda monitored by the knock-off method for areas A and B. In the second-flush tea season, the number of *E. onukii* Matsuda was almost five times higher in area B than in A, suggesting

that its occurrence varied between area A and B. The second-flush tea leaves, which were most affected by insect feeding, were used for the analysis of nonvolatile metabolites.

Analyses of catechins, TFs, and caffeine, which were characteristic components in tea leaves, were first performed (Figure 2). The caffeine content of leaves from area B was observed to be almost 1.2 times higher than those from area A. It has been reported that mRNA encoding caffeine synthesis is increased by insect feeding [36]. This suggests that caffeine biosynthesis is induced by insect feeding, and our results are consistent with this report.

The levels of catechins in area A were generally consistent with the significant effects of seasonal variations on catechin concentrations reported in earlier studies [9,37,38]. Liao et al. investigated the influence of insects on metabolites that affect tea quality, such as catechins. Their study was conducted using the model system in which *Ectropis grisescens* Warren or *E. onukii* Matsuda were introduced to potted tea plants in mesh bags [12]. They reported that 96 h of insect treatment did not affect the levels of catechins. Their results are consistent with our findings, in which no difference in total catechins was observed between areas A and B. Interestingly, in second-flush tea leaves, differences in catechin composition were observed in area B, especially an increase in gallate-type catechin. On the other hand, no difference in compositions of catechins were found in first-flush and third-flush tea leaves, which had less insect infections. EGC is biosynthesized by the flavonoid pathways, and a gallate group is attached to produce EGCG by epicatechin:1-O-galloyl-β-d-glucose O-galloyl transferase [39]. The difference in catechin compositions suggested an enhanced conversion of EGC to EGCG. Because gallate-type catechins have higher antibacterial activity than non-gallate-type catechins [40], the quantity of EGCG might be increased as a defense response. The results of catechin quantification suggested that the total catechin quantity was more influenced by the season, and the catechin composition was more influenced by insect feeding.

No significant changes in TFs were observed in this study, differing from the previous study by Liao et al. [12]. TFs are oxidative catechin dimers, which are formed when catechins in the vacuole mix with polyphenol oxidase due to leaf damage, such as those caused by insect feeding. A comparison study of intact and mechanically damaged leaves reported that the TF content of mechanically damaged leaves was higher than that of intact leaves [12]. This suggests that TF levels are strongly influenced by the degree of direct leaf damage rather than by insects.

OPLS-DA (Figure 3) and molecular network (Figure 4) analysis were applied to determine the characteristic metabolites induced by insect damage. Afzelechin gallate, procyanidins, hydrolyzable tannin, and monoterpene glycosides were detected as characteristic components in area B (Table 1). Increasing afzelechin gallate was presumably due to an increase in the percentage of gallate-type catechins, similar to an increase in EGCG and ECG by insect feeding. Procyanidins and hydrolyzable tannins are reported as defensive substances against herbivores [41,42]. Additionally, the gene encoding the leucoanthocyanidin reductase enzyme, which is involved in the first step of procyanidin synthesis, is known to be upregulated by leafhopper treatment [36]. Thus, procyanidins and hydrolyzable tannins may have increased as a defensive response to insects.

Increasing volatile compounds, such as linalool, have been reported in tea leaves fed on by leafhoppers [17]. Monoterpenes, such as linalool and geraniol, are defensive substances in plants [36]. Moreover, volatile compound glycosides are known to be in storage form [43]. In this study, glycosides of volatile compounds in area B were abundant because these compounds may have been accumulated by insect feeding.

Next, the effects of different insect damage levels in fresh leaves on black tea were examined. Some of the compounds mentioned in fresh tea leaves were also detected as characteristic components in the teas from area B. Therefore, the difference in components in fresh tea leaves directly affected the components of black tea products. Moreover, theaflavin-3-O-gallate and theaflavin-3,3′-O-gallate contents in tea leaves from area B

were higher than those in area A (Figure 5). These results were attributed to the higher gallate-type catechins of the fresh leaves in area B.

There was a color difference in the black tea products from each area (Table 3). Tea components are changed drastically by enzymes during processing. TFs and thearubigin are known as the dominant color components in black tea and are mainly formed from catechins [22,27–29]. Therefore, it is possible that differences in the composition of the fresh leaves, especially catechin, affected the color of black tea.

## 5. Conclusions

In this study, tea leaves grown in fields in moderately insect-attacked (area A) and insect-attacked areas (area B) were analyzed, with a particular focus on nonvolatile components. Among the catechins, gallate-type catechins increased in area B. In addition to catechins, increased procyanidins and monoterpene glycosides were found in area B by combining OPLS-DA and MN analysis. The results of this study differ from some of the findings of laboratory-based insect feeding models. This may be due to a synergistic effect of various insect species in the actual garden, not just specific model insects. Some of these compounds were also detected as characteristic components in black tea prepared from tea leaves of area B. This result suggested that changes in the components of an insect attack affect the quality of black tea products. Moreover, the color of the tea infusion is an important parameter in evaluating the quality of black tea. The infusion color of the area B leaves was darker and more orange than those from area A. Since black tea pigments are formed from the components of fresh leaves during processing, the component changes in the fresh leaves may affect the black tea color.

The effect of insect feeding on tea components was greater than expected. Insect damage is an environmental factor that cannot be ignored in evaluating tea quality. However, polyphenols such as procyanidins and gallate-type catechins, which were found to be increased by insect damage, have functional properties such as antioxidant activity. In other words, moderate insect damage has the potential to enhance tea functionality. Further research on optimal insect pest management is required to produce higher quality and more functional tea leaves in the future.

**Supplementary Materials:** The following supporting information can be downloaded at: https://www.mdpi.com/article/10.3390/horticulturae9101078/s1, Figure S1: Calibration curves for catechins, theaflavins, and caffeine; Figure S2: The study design and methodology; Table S1: List of IDs detected by MZmine; Figure S3: PCA based on the UPLC–MS spectra of the second-flush tea leaf extracts. (A) PCA score plot, and (B) loading plot; Table S2: Component abundance in area B (second-flush fresh tea); Figure S4: PCA based on the UPLC–MS spectra of the second-flush tea leaf extracts; Figure S5: OPLS-DA based on the UPLC–MS spectra of the black tea leaf extracts. (A) OPLS-DA score plot, and (B) S-plot; Table S3: Component abundance in area B (second-flush black tea); Figure S6: The structures of catechins, caffeine, and procyanidin.

**Author Contributions:** A.I. performed the experiments and analyzed the data. A.I. and E.Y. wrote the manuscript. J.K., N.K. and E.Y. designed the study. E.Y. supervised the study. All authors have read and agreed to the published version of the manuscript.

**Funding:** This work was supported by a JSPS Kakenhi Grant (21K05421).

**Data Availability Statement:** The datasets generated and analyzed during this study are available via MassIVE, at https://doi.org/doi:10.25345/C5W37M58T (accessed on 21 July 2023) under the MSV000092494.

**Acknowledgments:** We would like to thank Munenori Yagyu (Ibi Region Agriculture and Forestry Office, Gifu Prefecture, Japan) and Kenji Hamasaki (Gifu Ken Byogaityu Bozyo Center, the pest control center of Gifu Prefecture, Japan) for monitoring the number of insects.

**Conflicts of Interest:** The authors declare no conflict of interest.

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
