# Peer review of "Alterations in Nonvolatile Components of Tea (Camellia sinensis) Induced by Insect Feeding under Field Conditions"

_horticulturae, doi:10.3390/horticulturae9101078_

Round 1

Reviewer 1 Report

Dear Authors, I congratulate you on the manuscript, but the manuscript needs to be modified. I will explain this below:

The Introduction is short and rough, superficial.  It would be worthwhile to supplement it with literature related to tea pests or the tea plant. I think that the literature on the subject is much more extensive than you have put these few lines together. Please write again.

There is certainly literature on the measurements described in the method, with which the chapter can be expanded. You don't need to delete anything, but you must complete the chapter with the above descriptions. 

Where is the Discussion chapter? It is combined with the Results and thus difficult to understand. Not good, it would be better to include this in a separate chapter.

The objectives are not clearly stated.

The Conclusion is short and rough. Please rewrite.

How are the References numbered? Why are the numbers listed twice?

Rough work, but the topic is good. Please make major changes.

The English language is adequate, but minor modifications (stylistics) are necessary.

Reviewer 2 Report

General comment:

This paper study about the non-volatile components that contribute to a plant’s defence by setting up insect-attacked and moderately insect-attacked areas in the field to investigate the effect of insect feeding on non-volatile components. Overall, the paper was written in an easy understanding manner. However, there were still some points that need to be revised as it remains confusing for the readers. To conclude, this paper needs to revise it carefully before it can be considered in high impact journal. Hope below comments will able to help to further improve the paper.

Specific comment:

Abstract:

-      Suggest to modify the title to be more attractive and related to the current trends.

-      Needs major revision prior to the amendment of the main content.

-      An abstract is often presented separately from the article, so it must be able to stand alone. Hence the problem statement, aim, novelty and results of the study, all should be included into the one paragraph of abstract.

Introduction:

-      Introduction should be covered the gap of the research. However, it is not well covered in this section.

-      Also, please mention the important of this study to society as well as industry.

-      Problem statement of your introduction is not strong, need to discuss more about it.

-      Articles should be written in a passive manner. Avoid using the term ‘we’. Please revise.

-      The objective of the research remains unclear. Please revise.  

-      Revised Introduction section based on the structure below:

1st paragraph: Problem statement

2nd paragraph: Current ongoing solution

3rd paragraph: Proposed solution in this work.

4th paragraph: Summarized the current research novelty and objective of this work.

-      Kindly refer some latest papers as it is highly relevant to this report. Example, Bridge between mass transfer behavior and properties of bubbles under two-stage ultrasound-assisted physisorption of polyphenols using macroporous resin; Thermo-sensitive aqueous biphasic extraction of polyphenols from Camellia sinensis var. assamica leaves

Material and Method:

-      Please cite relevant references for sample preparation and UPLC where the procedure was adopted from.

-      Please provide an additional figure to illustrate the process of the whole methodology.

Results and discussion:

-      Is there any reason why the insect distributes this way? Due to season or due to plant growth and mature? Please specify in the manuscript.

-      Line 199-201; “In area A, second-flush tea leaves had a higher level of catechins than first-flush tea leaves and the same level as third-flush tea leaves…” please revise the sentence as it is confusing. From the figure second and third flush of tea shared the same number instead of with first flush. Please revise.

-      Line 245; please provide full name of the term ‘VIP’ before using abbreviation.

-      Figure 5; is there any meaning for the wavelike shape object placed on the figure? If so, what does it mean?

-      Figure 3&4 showed in the paper is in low resolution. Please provide higher quality of figures and images.

-      The overall structure needs to be improved. Avoid one sentence paragraph. Try merging it into the paragraph before or after.

-      Authors should pay attention to English grammar and sentence structure. The authors are suggested to edit the manuscript carefully before submission.

-      The authors are encouraged to read this article for more information, Effects of edible coatings of chitosan-fish skin gelatine containing black tea extract on quality of minimally processed papaya during refrigerated storage; Development of Edible Coating from Gelatin Composites with the Addition of Black Tea Extract (Camellia sinensis) on Minimally Processed Watermelon (Citrullus lanatus); Effect of major tea insect attack on formation of quality-related nonvolatile specialized metabolites in tea (Camellia sinensis) leaves

Conclusions

-      Good but please include the limitations and what can be done for the future studies.

References

-      Kindly revise reference format according to the author guideline.

Extensive editing of English language required

Reviewer 3 Report

Ayumi Ito et al. analyzed tea from moderately infested areas (zone A) and infested areas (zone B), focusing on non-volatile components in tea. The orthogonal partial least square discriminant analysis and molecular network analysis of tea were carried out by ultra-high performance liquid chromatography-mass spectrometry. The results show that it is not always consistent with the results of the laboratory insect feeding model system, which validates the importance of pest control in maintaining the quality of tea, but there are still some major points that need to be addressed.

1. In line 46, according to the literature, the principle and formation of the specific model are briefly introduced, which is convenient to compare with the actual environment.

2. The Data Analysis section should be short and highlight the main parts

3. In line 46, Citing previous research by othersit was proved that its metabolites were inconsistent with the model.

4. Add statistics and analysis at the end of Materials and Methods.

5. The ordinate units in Figure1 are missing.

6. In Figure1, the time selection ranges of (A) and (B), (C) and (D) are different. What is the basis for this.

7. In line 222, replace "have been reported" with "has been reported".

8. In Table 3, it is suggested to make three parallel sets for the determination of color difference to increase the credibility.

9. Duplicate numbering before reference should be avoided

Reviewer 4 Report

In this research, the authors primarily focused on examining the effect of insect feeding on catechin, theaflavins and other nonvolatile compounds of tea leaves and black tea as a product. The topic is interesting, and the analyzes are adequately selected and comprehensive. In its current form, the manuscript is quite good and has minor flaws, which I suggest the authors correct:

Line11; Unclear sentence, rephrase it.

In the abstract, state some concrete result obtained, for example, how many times higher is the content of certain nonvolatile compounds in the area B than in area A.

Line 46-48; Write in more detail about the model systems that were used in the experiments and the results  that accompany them.

Line 67-69; Why pesticides were used in area A and not in B, and how it is comparable afterwards?

Line 81; Write what species of plant the leaf samples are.

Line 83; Type and origin of the lyophilizer used?

Figure 1 (A) comment on the infestation of insects in areas A and B and why the comparison of the two areas is not represented in the other graphics (B, C and D).

Line 208 and 225; The reference number is missing.

Line 334-358; Compare with the conclusions and results of other authors who investigated a similar field of research.

Give a suggestion for future research in the conclusion.

Round 2

Reviewer 1 Report

Dear Authors,

I accept the manuscript for publication.

Reviewer 2 Report

The manuscript is corrected and revised according to the reviewer's comments. I am now satisfied with the new version, so I would like to recommend its publication.

Minor editing of English language required

Reviewer 3 Report

All issues were addressed.